# Nodule Synthetic Bacterial Community as Legume Biofertilizer under Abiotic Stress in Estuarine Soils

**DOI:** 10.3390/plants12112083

**Published:** 2023-05-24

**Authors:** Noris J. Flores-Duarte, Salvadora Navarro-Torre, Enrique Mateos-Naranjo, Susana Redondo-Gómez, Eloísa Pajuelo, Ignacio D. Rodríguez-Llorente

**Affiliations:** 1Departamento de Microbiología y Parasitología, Universidad de Sevilla, 41012 Seville, Spain; nflores@us.es (N.J.F.-D.); epajuelo@us.es (E.P.); 2Departamento de Biología Vegetal y Ecología, Universidad de Sevilla, 41012 Seville, Spain; emana@us.es (E.M.-N.); susana@us.es (S.R.-G.)

**Keywords:** abiotic stress, legumes nodulation, plant growth promoting endophytes, SynCom, degraded estuarine soils

## Abstract

Estuaries are ecologically important ecosystems particularly affected by climate change and human activities. Our interest is focused on the use of legumes to fight against the degradation of estuarine soils and loss of fertility under adverse conditions. This work was aimed to determine the potential of a nodule synthetic bacterial community (SynCom), including two *Ensifer* sp. and two *Pseudomonas* sp. strains isolated from *Medicago* spp. nodules, to promote *M. sativa* growth and nodulation in degraded estuarine soils under several abiotic stresses, including high metal contamination, salinity, drought and high temperature. These plant growth promoting (PGP) endophytes were able to maintain and even increase their PGP properties in the presence of metals. Inoculation with the SynCom in pots containing soil enhanced plant growth parameters (from 3- to 12-fold increase in dry weight), nodulation (from 1.5- to 3-fold increase in nodules number), photosynthesis and nitrogen content (up to 4-fold under metal stress) under all the controlled conditions tested. The increase in plant antioxidant enzymatic activities seems to be a common and important mechanism of plant protection induced by the SynCom under abiotic stress conditions. The SynCom increased *M. sativa* metals accumulation in roots, with low levels of metals translocation to shoots. Results indicated that the SynCom used in this work is an appropriate ecological and safe tool to improve *Medicago* growth and adaptation to degraded estuarine soils under climate change conditions.

## 1. Introduction

Estuaries are complex ecosystems that provide material and nonmaterial benefits to humans [1,2]; however, estuaries also suffer human disruption and overexploitation, resulting in their impoverishing [3]. Climate change affects these coastal ecosystems by increasing sea surface temperature, heat (or cold) waves and frequency of extreme events, such as strong reduction in rainfall (and consequently, drought events) [4]. Modification of water quality, migration patterns and plant growth are also among the effects of climate change on estuaries [5]. In that way, the combined effects of human practices, particularly industrial and agricultural activities, and climate change have contributed to increased estuarine soils degradation, limiting plant development and diversity. Vegetal cover plays a crucial role in estuarine ecosystems conservation, since plants reduce soil compaction and erosion, increase the content of organic matter and microorganisms’ biodiversity and improve soil hydraulic and structural properties [6]. Furthermore, recovery of degraded estuarine soils implies revegetation, preferably using autochthonous plant species that should include legumes. Legumes can fix atmospheric nitrogen through their interaction with soil bacteria commonly named rhizobia [7], improving soil health and fertility and providing a large proportion of the nitrogen requirements for plants [8].

One example of degraded estuarine ecosystem is the joint estuary of the Odiel and Tinto rivers in Huelva region (SW Spain). Due to both natural and anthropogenic activities, this estuary became one of the most polluted areas in the world [9]. Levels of toxic metals—such as As, Cu, Pb or Zn—exceeding those allowed by regional and national legislation in natural parks, agricultural and industrial soils [10] has been recorded in both Odiel and Tinto river marshes in the last decade [11,12]. These levels of soil contamination make their recovery mandatory. Several works using autochthonous halophytes and their associated rhizospheric and endophytic bacteria aimed to alleviate the ecological effects of metal contamination in this estuary have been reported. These publications evaluated the increase of the phytoremediation potential of *Spartina maritima* [13,14], *Arthrocnemum marostachyum* [12] or *Salicornia ramosissima* [15] in estuarine soils bioaugmented with bacteria from their microbiomes. In general, the selected bacteria enhanced plant growth in presence of metals and increased metal accumulation in roots, improving the phytostabilization potential of the halophytes. Bacteria characterized and used in these works showed several plant-growth promoting (PGP) properties, such as phosphate solubilisation, nitrogen fixation or auxins and siderophores production, among others, that could be responsible of the effects observed in plant growth [16].

More recently, the potential use of legumes, particularly *Medicago sativa*, combined with both rizospheric and endophytic bacteria to improve estuarine soils fertility has been investigated [17,18]. In both works, inoculation with selected bacteria resulted in increasing *M. sativa* growth and nodulation, ameliorating plant photosynthesis and physiological parameters. In this context, we have described four endophytes synthetic bacterial community (SynCom), isolated from *Medicago* spp. nodules of plants naturally growing in Odiel river high marshes soils, capable of promoting *M. sativa* growth, nodulation and phytostabilization ability in metals contaminated estuarine soils [19]. This SynCom included two nodule inducing *Ensifer* sp. strains and two *Pseudomonas* sp. nodule endophytes with multiple PGP properties.

The aims of this work were to evaluate the ability of the *Pseudomonas-Ensifer* SynCom to enhance *M. sativa* growth and nodulation rates under high metal, salt, high temperature or drought stress in estuarine soils, in an attempt to fight against the present and future consequences of climate change and human activities on the soil fertility of Odiel river marshes.

## 2. Results

### 2.1. SynCom Characterization under Abiotic Stress

In a previous work, we isolated and characterized endophytes inhabiting inside nodules of *Medicago* spp. growing in degraded estuarine soils. A synthetic bacterial community (SynCom) was designed based on bacterial PGP properties, lytic enzymatic activities and tolerance to heavy metals (Appendix A) [19]. The SynCom was made up of four strains: *Pseudomonas* sp. N4, *Pseudomonas* sp. N8, *Ensifer* sp. N10 and *Ensifer* sp. N12 [19]. In this study, the PGP properties and the lytic enzymatic activities of these strains were tested in the presence of heavy metals (Table 1). All strains maintained the PGP properties in absence of metals, with the exception of strains N10 and N12, which did not grow in a minimal medium without a nitrogen source in presence of metals. It should be noted that biofilm capacity was present in all strains, but the absorbance decreased deeply in presence of heavy metals (Table 1). Interestingly, the ACC deaminase activity of the N4 and N8 strains increased in the presence of heavy metals, while N10 and N12, which did not show the ACC deaminase activity in the absence of metal, showed activity in their presence (Table 1).

Regarding the lytic enzymatic activities, all strains showed pectinase activity in the presence of heavy metals, while not in their absence (Table 1). The N10 and N12 maintained the DNAse activity but lost the ability to degrade cellulose in the presence of metals (Table 1). On the other hand, strains N4 and N8 only showed pectinase activity in the presence of metals, losing both proteinase activity and N8 cellulase activity (Table 1).

In addition to these properties, the ability of these bacteria to grow under saline conditions was tested. All strains tolerated high concentrations of NaCl reaching values of 1.5 M in the case of strains N4 and N8, and 2 M for strains N10 and N12 (Table 1).

Regarding tolerance to drought, bacterial strains were exposed to different concentrations of polyethylene glycol (PEG) since this polymer simulates the drought conditions. Strains N4, N10, and N12 were able to grow at 30% of PEG, and strain N8 at 25% PEG. Finally, all strains tolerated temperatures similar to or greater than 40 °C (Table 1).

### 2.2. Inoculation with the SynCom Enhances M. sativa Growth under Stressful Conditions

The utility of the designed SynCom to promote the growth of *M. sativa* under high metal stress was tested using sterilized soil from the low marshes in the Odiel river estuary. Metal concentrations in this soil are presented in Table 2.

The inoculated plants were significantly taller and leafier than the non-inoculated ones, showing that the SynCom up to three-fold the length and five-fold shoot dry weight of the shoot compared to non-inoculated plants (Figure 1A,B). Similar results were observed in roots comparing non-inoculated and inoculated plants, although those plants inoculated with the SynCom showed a similar length of roots than those inoculated with *Ensifer* sp. N10 alone (Figure 1B). However, the roots of the plants inoculated with SynCom were thicker, with more secondary roots and root hairs than the plants inoculated with strain N10 (Figure 1A). The number of nodules was significantly higher in plants inoculated with SynCom (Figure 1C) and the total N content was also higher—close to 300% more N than in non-inoculated plants (Figure 1D).

To assess the effect of SynCom inoculation under salinity stress, sterilized soil from the upper marshes of the Odiel river was supplemented with sterile NaCl solution to reach a final NaCl concentration of 60 mM in the soil, as described in Section 4.

Plants inoculated with *Ensifer* sp. N10 significantly improved their biomass and length compared to non-inoculated plants (Figure 1A,B). A higher improvement was measured in plants inoculated with SynCom, increasing the shoot biomass by 2300% compared to non-inoculated plants (Figure 1A). However, the SynCom-inoculated plants showed the lowest values of root length, although the roots were thicker and presented more lateral roots than in the other two conditions, as reflected in root biomass (Figure 1A,B). Furthermore, SynCom ameliorated plant nodulation, showing almost three times the number of nodules counted in plants inoculated with N10 (Figure 1C). The total nitrogen content in plants also increased in presence of bacteria and the highest amount was observed in plants inoculated with SynCom (Figure 1D).

Regarding *M. sativa* growth and nodulation under drought stress, this abiotic stress particularly affected the development of plant shoots (Figure 1A). Again, plants inoculated with the SynCom showed the highest dry weight and length values of the roots and shoots, while plants inoculated with the *Ensifer* strain reported higher values than non-inoculated plants (Figure 1A,B). Although SynCom significantly enhanced *M. sativa* nodulation and plant nitrogen content (Figure 1C,D), nitrogen content was particularly affected by this stress condition.

Finally, the response to *M. sativa* growth after a heat shock and the effect of bacterial inoculation in plant protection were evaluated. The high temperature affected the development of the shoot in longer extension than the development of roots (Figure 1A,B). Inoculation with the SynCom significantly improved root and shoot development. Plants inoculated with SynCom under heat stress showed the longest roots and the highest dry weight compared with the other stress conditions. Concerning nodulation, although inoculation with SynCom increased nodule number, the temperature particularly affected nodulation, showing these plants the lowest number of nodules among plants under the different stress conditions (Figure 1C). However, the nitrogen content was similar to those recorded in plants under salt or heavy metal stress (Figure 1D).

### 2.3. Inoculation with the SynCom Enhances M. sativa Photosynthetic Parameters under Stressful Conditions

In addition to the improvement in plant growth and nodulation, the bacterial SynCom also improved the photosynthetic parameters in a statistically significant way, compared with plants not inoculated or inoculated with *Ensifer* sp. N10 in the presence of all the abiotic stress conditions (Figure 2). Under high concentrations of heavy metals, the net photosynthetic rate showed the greatest improvement after inoculation with SynCom (Figure 2B).

Plants treated with 60 mM of NaCl and inoculated with SynCom showed increases over 100% in total chlorophyll content, PSII efficiency, iWUE and electron transport rate compared with single inoculated or non-inoculated plants (Figure 2A,D–F).

Under drought stress, the total chlorophyll content and iWUE significantly increased their values in SynCom-inoculated plants (Figure 1A,E).

On the other hand, high temperatures particularly affected plant photosynthesis (Figure 2). Inoculation with the SynCom increased all the parameters measured, but the values of these parameters were, in general, clearly lower than the values observed in plants under the other stress conditions. In particular, the net photosynthetic rate (A_N_), the quantum efficiency of PSII (Φ_PSII_) and the electron transport efficiency (ETR) showed very low values under this condition (Figure 2B,D,F).

### 2.4. Inoculation with the SynCom Enhances M. sativa Stress Status under Stressful Conditions

In this work, the stress level in plants was also checked by measuring antioxidant enzymes activities. Inoculation with the SynCom increased antioxidant enzyme activities over the values recorded in non-inoculated or single-inoculated plants (Figure 3). Under heavy metal stress, this enzymatic induction was also observed in plants inoculated with strain N10 compared to non-inoculated plants, but with lower differences (Figure 3). Enzymatic activities were higher in roots than in shoots (Figure 3). In particular, catalase and ascorbate peroxidase activities were not significantly different in shoots of non-inoculated and single-inoculated plants (N10) (Figure 3C,D).

Concerning salinity stress, these enzymatic activities were more pronounced in roots than in shoots, where antioxidant enzymes showed higher activities (Figure 3). In general, inoculation with bacteria induced the activity of these plant enzymes, and the highest values were again recorded in plants inoculated with SynCom (Figure 3).

A similar pattern to that described in plants under metal stress was observed in plants exposed to a drought and a high temperature period. The activities of antioxidant enzymes increased in plants inoculated with the bacterial community, although higher values of activity were observed in roots than in shoots in the case of drought (Figure 3).

### 2.5. Inoculation with the SynCom Modifies M. sativa Metals/loids and Micronutrients Accumulation under Heavy Metal and Salinity Stress

Finally, the concentration of the most abundant and toxic metal/loids was measured in roots and shoots in order to determine their accumulation in plant tissues when plants were grown in soils with high concentrations of heavy metals. As shown in Figure 4, inoculation with SynCom promoted As, Cd, Cu and Zn, with significant differences compared to the metals accumulated in non-inoculated or single-inoculated plants. These differences in metal/loids accumulation were greater in roots than in shoots and, in all plants, the accumulation was much higher in roots than in shoots.

In plants treated with 60 mM NaCl, the concentration of elements such as K, Mn, Na and P was higher in plants inoculated with the SynCom than in control plants (Appendix A).

## 3. Discussion

In the last decade, our efforts have been focused on the recovery of the degraded estuarine soils of southwest Spain using autochthonous halophytes and their associated bacteria [11,12,13,14,15]. Recently, we have proposed the combined use of rhizobia and plant growth promoting nodule endophytes to inoculate autochthonous *Medicago* spp. plants in an attempt to recover degraded estuarine soils by increasing their fertility and reducing metals impact in plants [19]. We selected an endophytic bacterial SynCom containing two *Ensifer* sp. and two *Pseudomonas* sp. strains with several PGP properties that were able to promote *M. sativa* growth and nodulation on degraded estuarine soils. *M. sativa* is a crop forage plant widely used in Spain for centuries and, although it is not a native species, it is a naturalized one. Our purpose in this work was to further elucidate the utility of the designed nodule SynCom to enhance *M. sativa* adaptation and growth in estuarine soils under a plethora of stresses in a climate change scenario. Odiel river estuarine (SW Spain) is located in a region that experiences long periods of drought, in which temperatures can reach 40 °C at central hours of the day during summers, under a heavy sun. The future prospective is even worse, with longer periods of drought and high temperatures.

*M. sativa* plants subjected to abiotic stressing situations increased their growth, nodulation rate compared with single inoculated plants and improved their photosynthetic parameters in response to inoculation with our nodule SynCom. These results are related with the presence in these endophytes of PGP properties. IAA production is involved in the increase of roots and shoots length [20], since this hormone increase cell expansion and is essential in root development [21]. Bacterial properties related with plant nutrient acquisition also influenced *M. sativa* growth. Phosphate solubilization provided phosphorus [22,23], which helps in the nodulation process, amino acid and proteins synthesis [24]. Siderophores produced by bacteria could provide essential metals such iron and Zn [25]. Concerning nitrogen, our SynCom enhanced *M. sativa* nodulation thus increased nitrogen content in plants. Increase in nodulation rate could be related with ACC deaminase activity, which facilitated the reduction of the stress levels in the plants by reducing ethylene and then favoring nodulation [26]. Ethylene accumulation is also responsible for defoliation, cell elongation inhibition and senescence [27]. IAA also interferes in the rhizobia-legume dialogue interaction, delaying nodule senescence by interacting with the bacteroid inside the nodules [28]. *Ensifer* sp. and *Pseudomonas* sp. strains showed ACC deaminase activity and the ability to produce auxins, solubilize phosphate and form biofilms in the presence of As, Cd, Cu and Zn. Interestingly, the presence of metals increased the levels of ACC deaminase activity in *Pseudomonas* sp. strains and induced this activity in *Ensifer* sp. strains, that lacked it in absence of metals. It is important to note that, despite one of the stress conditions, was the high metal content in soil—the soil from the estuarine used in this work to assay the other stresses had moderate levels of metal contamination.

The positive effect of inoculation with the SynCom on *M. sativa* growth and nodulation under abiotic stress was also reflected in increased values of photosynthesis parameters. Nevertheless, plants growing at high temperature showed values particularly low of parameters such as net photosynthetic rate (A_N_), the quantum efficiency of PSII (Φ_PSII_) and electron transport efficiency (ETR), even under SynCom inoculation. There are several reviews reporting the effects of high temperature in photosynthesis, which is highly sensitive to this stress [29]. Among others, high temperature inhibits PSII, which efficiency was partially restored by SynCom inoculation in this work (Figure 2D). Heavy metal and drought stresses particularly affected the net photosynthetic rate, while very low values of iWUE were recorded in control non-inoculated plants under salt stress. Values of iWUE under salt stress were completely restored in plants inoculated with the SynCom (Figure 2E). In this work, *M. sativa* plants inoculated with the SynCom increased their chlorophyl content under the different stresses. Concerning legumes, enhanced amounts of chlorophyl were also found in *M. sativa* plants inoculated with two *Variovorax* endophytes under metal stress [18], in chickpea plants inoculated with the endophyte *Bacillus subtilis* BERA 71 under salt stress [30] or in soybean plants inoculated with the endophyte *Bacillus cereus* SA1 under high temperature [31]. *Alyssum serpyllifolium* plants inoculated with *Pseudomonas azotoformans* ASS1, a drought-resistant endophyte, showed increased values of chlorophyl under drought stress [32].

It is well-known that production of an excess of reactive oxygen species (ROS) by plant and a decrease in the antioxidant defense mechanisms severely damage plants, altering cell function and reducing development [33,34]. Plant antioxidant defense enzymes, such as catalase, guaiacol peroxidase, ascorbate peroxidase and superoxide dismutase, act as protectors against ROS. In this work, activities of these enzymes increased in roots and shoots of *M. sativa* plants inoculated with the SynCom under the different abiotic stresses. These results suggest that bacterial induction of plant antioxidant enzymatic activities is a common and important mechanism of plant protection in abiotic stress conditions. *M. sativa* plants inoculated with rizospheric bacteria under heavy metal stress [35] or nutrient deficiency [17] showed increased levels of catalase, guaiacol peroxidase, ascorbate peroxidase and superoxide dismutase. Similar results were also found in *M. sativa* plants growing in metal contaminated estuarine soils inoculated with nodule endophytes from *Variovorax* genus [18]. Inoculation of chickpea plants with the endophytic bacterium *Bacillus subtilis* BERA 71 under saline conditions enhanced superoxide dismutase and catalase activities in the plant [30]. Soybean plants inoculated with the endophyte *Bacillus cereus* SA1 showed increased levels of ascorbate peroxidase and superoxide dismutase under high temperature stress [31]. In the same way, the drought-resistant endophytic bacteria *Pseudomonas azotoformans* ASS1 enhanced the activities of *Alyssum serpyllifolium* superoxide dismutase, peroxidase and catalase in inoculation experiments under drought stress [32].

Legumes are known to be able to accumulate high amounts of heavy metals in roots, with low levels of translocation to shoots, without affecting plant growth [36,37]. In this work, *M. sativa* plants inoculated with the SynCom in an estuarine soil with high levels of metals increased the amounts of As, Cd, Cu and Zn accumulated in roots, compared with non-inoculated plants or single inoculated with rhizobia. The concentration of metals in roots were similar to those measured in *M. sativa* roots of plants growing in an estuarine soil with moderate metals contamination and inoculated with a bacterial consortium containing two *Variovorax* species [18] or the SynCom used in the present work [19]. It is important to notice that metals content in shoots were below the values allowed for human or animal consumption, whose limits are 30 ppm As, 10 ppm Cd, 40 ppm Cu and 500 ppm Zn [10,19]. These results are in consonance with previous works indicating that nitrogen-fixing bacteria and endophytes inside nodules reduce metal translocation to legume shoots in metal contaminated soils and increase the phytostabilization potential of the plant [19,38].

These results suggest that we could use *M. sativa* inoculated with our SynCom to recover estuarine degraded soils and revegetate soil areas with high metal content or salinity, improving soil quality. This SynCom could also be an appropriate tool to fight against climate change in these ecologically important ecosystems, promoting legume adaptation. The SynCom could be used to inoculate autochthonous *Medicago* spp. plant naturally growing in estuaries in order to promote their growth in such soils. Although in situ experiments need to be done, inoculation with this SynCom seems to be a safe tool, since low amounts of metals are translocated to plant shoots.

## 4. Materials and Methods

### 4.1. Bacterial Strains Characterization

*Pseudomonas* sp. N4, *Pseudomonas* sp. N8, *Ensifer* sp. N10 and *Ensifer* sp. N12 strains were isolated from nodules of *Medicago* spp. plants naturally growing in high marshes of the Odiel river [19]. *Pseudomonas* strains were maintained in TSA (Tryptic Soy Agar; Intron Biotechnology, Seongnam, Republic of Korea) plates and *Ensifer* strains in TY (Tryptone-Yeast extract) [39] plates. The growth conditions were 24 h at 28 °C.

#### 4.1.1. Characterization of Plant Growth Promoting Properties in presence of Heavy Metals

Phosphate solubilization, indole-3-acetic acid (IAA) production, biofilm formation, ACC deaminase activity and nitrogen fixation were assayed in the presence of heavy metals. A mix solution of heavy metals was created from sterile stock solutions of 1 M NaAsO_2_, 1 M CdCl_2_, 1 M CuSO_4_ and 1 M ZnSO_4_, which were added to the different media after autoclaving to reach a final concentration of 0.3 mM of As, Cu and Zn and 0.05 mM of Cd. Methods to determine each property are described in [19]. Briefly, phosphate solubilization was checked by the appearance of transparent halos around the bacterial growth in NBRIP (phosphate growth medium from the National Institute of Botanical Research) plates (tricalcium phosphate as substrate) after 7 days at 28 °C [40]. IAA production was determined by measuring the absorbance at 535 nm after the addition of Salkowski’s reagent [41] to the supernatant of a 24 h culture in TSB (Tryptic Soy Broth; Intron Biotechnology, Seongnam, Republic of Korea) or liquid TY supplemented with L- Tryptophan (100 mg·L^−1^). IAA concentration was determined using a pattern curve. The capacity to form biofilm was examined in TSB or liquid TY using a 96-wells plate after 4 days of incubation, with a previous stain with 0.01% violet crystal. Then, staining cells were resuspended adding acetic acid/ethanol and the absorbance was measured at 570/595 nm [42]. To determine the ACC deaminase activity, the formation of α-ketobutyrate was measured at 540 nm using a spectrophotometer (Lambda25; PerkinElmer, Walthmam, MA, USA) following the protocol described by Penrose and Glick [43] and later adapted [11]. For this, bacteria were growing in the presence of ACC for 24 h, and the total protein content were then extracted using toluene. α-ketobutyrate concentration was determined using a pattern curve and the ACC deaminase activity was expressed in μmoles of α-ketobutyrate per mg of protein per hour. The nitrogen fixation was studied as first approximation by growing the strains in NFB (Nitrogen Free Broth) plates for 5 days at 28 °C [44].

#### 4.1.2. Characterization of Enzymatic Activities in Presence of Heavy Metals

DNAse, amylase, cellulase, lipase, pectinase, protease and chitinase activities were analyzed in the presence of heavy metals. The mix solution of heavy metals described above was added to the different media after autoclaving to reach a final concentration of 0.3 mM of As, Cu and Zn and 0.05 mM of Cd. Protocols are deeply described in [19]. Briefly, lipase, protease and chitinase activities were observed directly in plates with Tween80 [45], casein [45] and chitin [14] as substrates, respectively, with the appearance of a halo around the bacterial biomass. In a similar way, DNAse, amylase, cellulase and pectinase activities were checked in plates containing DNA (DNA agar medium; Scharlab, Barcelona, Spain), starch (Starch agar medium; Scharlab, Barcelona, Spain), carboxymetyl-cellulose and pectin as substrates, respectively [46]. However, in these cases, the results were observed indirectly after adding reveal solutions to plates such as 1N HCl (for DNAse), lugol (for amylase), 0.1% Congo red and 1M NaCl (for cellulase), and 2% CTAB (for pectinase). Plates to determinate these activities were incubated 7 days at 28 °C.

#### 4.1.3. Characterization of Tolerance against Abiotic Stress

Bacterial growth under NaCl, temperature and drought stress was tested. Tolerance against heavy metals was determined by Flores-Duarte et al. [19]. Tolerance to NaCl was studied by growing the strains in TSA or TY increasing NaCl concentrations, from 0.25 M to 2 M for 48 h at 28 °C. The maximum tolerable concentration (MTC) was determined for each strain. To determine growth at increasing temperatures (25, 30, 35, 40, 45 and 50 °C), strains were plated in TSA or TY and incubated for 48 h. Finally, the drought assay was performed using polyethylene glycol (PEG) because it is a natural polymer that simulates the drought conditions. This assay was performed in a 96-wells plates containing TSB or liquid TY with different concentrations of PEG (from 5% to 30%). Plates were incubated at 28 °C for 48 h. The MTC was determined for each strain.

### 4.2. Greenhouse Assays under Abiotic Stress

Four experiments to study different abiotic stresses were performed under greenhouse conditions using *Medicago sativa* plants: heavy metal pollution, salinity, high temperature and drought. Seeds of *M. sativa* were surface disinfected with ethanol and sodium hypochlorite as described in Flores-Duarte et al. [19], and then incubated in 0.9% water agar at 28 °C for 72 h in darkness. In total, 3-day-old seedlings were transferred to pots containing the corresponding substrate for each assay (2 seedlings per pot, 8 pots per treatment) and grew under greenhouse conditions for 60 days. Plants were inoculated every week with 50 mL of the corresponding inoculum (containing 10^8^ cells/mL). The inoculums were made mixing the bacterial culture (washing previously with sterile saline solution) with tap water according to the instructions detailed in Flores-Duarte et al. [19]. The inoculation treatments were: C- (without bacteria), N10 (inoculation with *Ensifer* sp. N10) and CSN (inoculation with *Pseudomonas* sp. N4, *Pseudomonas* sp. N8, *Ensifer* sp. N10, and *Ensifer* sp. N12).

#### 4.2.1. Heavy Metals Stress Assay

The soil used to observe the effect of high concentrations of heavy metals in plant growth was collected from the middle marshes of the Odiel river (Huelva, Spain; 37°150′ N, 6°580′ W) using gloves and a sterile spoon and it was transported to the laboratory in plastic bags. In the laboratory, this soil was analyzed according to Mateos-Naranjo et al. [47]. The texture was determined using the Bouyoucos hydrometer method [48]. The electrical conductivity was measured using a conductivity meter (Crison-522; Crison Instrument, S.A., Barcelona, Spain), and the redox potential and pH with a Crison pH/mVp-506 (Crison Instrument, S.A., Barcelona, Spain). Macro- and micronutrients concentrations were determined by inductively coupled plasma–optical emission spectroscopy (ICP–OES) (ARLFisons3410, Thermo Scientific, Walthman, MA, USA) and the organic material as described by Walkley and Black [49]. Before used as substrate in pots, the soil was sterilized by autoclaving three times. Plants were irrigated once a week with sterile water.

#### 4.2.2. Salinity Stress Assay

For this assay the soil was collected from the upper marshes of the Odiel river and analyzed as described above. It was sterilized three times by autoclaving and then distributed in pots. Pots for the same inoculation treatment were allocated in the same tray containing 3 l of a sterile 60 mM NaCl solution. This solution was changed every week.

#### 4.2.3. High Temperature Stress Assay

This assay was performed using autoclaved soils from the upper marshes of the Odiel river. Then, 41 days after the start of the experiment, plants were subjected to 40 °C during 5 consecutive days in a growth chamber. After this temperature shock, plants continue growing under greenhouse conditions for 2 more weeks for recovery. Plants were irrigated once a week, except during the temperature shock, when plants were irrigated every day.

#### 4.2.4. Drought Stress Assay

For the drought stress, sterilized soil from the upper marshes of the Odiel river was used as substrate. Plants grew for 25 days and then were exposed to drought stress for 3 weeks. Finally, the recovery phase lasted 2 weeks. During this assay, plants were irrigated once a week, except in the drought period. In that period, plants were non-irrigated and non-inoculated.

### 4.3. Plant Physiology Status Determination after Stressful Conditions

#### 4.3.1. Photosynthetic Parameters

In the heavy metal and salinity experiments, the photosynthetic parameters were measured after 60 days of plant growth. For drought and temperature stress assays these parameters were measured during the stress. The gas exchange was measured in random leaves from each plant between 10 a.m. and 2 p.m. under a photosynthetic photon flux density of 1500 µmol·m^−2^·s^−1^, a deficit of vapor pressure of 2–3 kPa, a temperature around 25 °C, and a CO_2_ concentration environment of 400 µmol·mol^−1^ air. Gas exchange stabilization (120 s) was equilibrated before measurements. For that, an infrared gas analyzer (IRGA) with a light chamber Li-6400-02B connected (LI-COR Biosciences, Lincoln, NE, USA) was used. With the recorded measurements, the net photosynthetic rate (A_N_) and the intrinsic water use efficiency (iWUE) were calculated. The efficiency of the energy used of the photosystem II (PSII) was also analyzed using a modulate fluorimeter FMS-2 (Hansatech Instruments Ltd., Pentney, UK). Following the protocol described by Schreiber et al. [50], selected leaves were dark and light adapted for 30 min, and then a saturating actinic light pulse of 10,000 μmol m^−2^ s^−1^ was given for 0.8 s at midday (1700 μmol photons m^−2^ s^−1^). Then, the maximum quantum efficiency of PSII photochemistry (Fv/Fm), the quantum efficiency of PSII (Φ_PSII_) were determined by a saturation pulse method [51] and the electron transport efficiency (ETR) was calculated with the recorded data [52].

Finally, random leaves were collected from each plant in order to extract the chlorophyll using acetone following the protocol described in Hiscox and Israelstam [53]. The absorbance of the extract was measured at 652 nm and the total chlorophyll content was calculated using the formula proposed by Arnon [54]. This measurement was performed in duplicate.

#### 4.3.2. Plant Growth Parameters and Nodulation

To determine the growth parameters, the length of shoots and roots was measured and the number of nodules was counted at the end of the experiments (after 60 days). Then, shoots and roots were dried in an oven, separately, at 80 °C for 2 days and the dry weight was determined.

#### 4.3.3. Analysis of the Antioxidant Enzymes

The activities of the plant antioxidant enzymes were analyzed following the protocol indicated by Duarte et al. [55]. In summary, the total protein content from leaves and roots was extracted from 500 mg of the vegetal material (previously frozen in liquid nitrogen) into extraction buffer (50 mM sodium phosphate buffer; pH 7.6) in ice using a homogenizator. The supernatant after centrifugation was stored at −80 °C until their use. Ascorbate peroxidase (APx) activity was measured mixing the vegetal extract with L-ascorbate, and the oxidation was recorded at 290 nm. For the superoxide dismutase (SOD), the autoxidation of the pyrogallol at 325 nm was measured. In the case of the catalase (CAT) activity, the H_2_O_2_ breakdown was monitored at 240 nm. Finally, the oxidation of guaiacol was observed at 470 nm to determine the activity of the guaiacol peroxidase (GPx). The enzymatic activities were related to the total protein content and expressed as units (U) per μg of protein. The amount of proteins extracted from each sample was analyzed using the Bradford method [56]. These measurements were performed in quintuplicate. These activities were determined after 60 days in plants treated with heavy metals and salt, and during the stress in plants exposed to drought and high temperature.

#### 4.3.4. Nitrogen, Heavy Metals and Salts Content Determination

The nitrogen content in shoots was determined using an InfrAlyzer 300 (Technicon, Tarrytown, NY, USA) from dry material [57]. For the heavy metals and salinity stress experiments, the content of As, Cd, Cu, Zn, Na, and K was measured by inductively coupled plasma–optical emission spectroscopy (ICP–OES) (ARLFisons3410, Thermo Scientific, Walthman, MA, USA).

### 4.4. Statistical Analysis

Data obtained in this study were checked to determine their normality using a Kolmogorov–Smirnov test. As all the data were normal, one-way ANOVA was used to compare them and the statistical differences were tested using the Fisher test. These analyses were carried out with Statistica software version 6.0 (Statsoft Inc., Tulsa, OK, USA).

## 5. Conclusions

In conclusion, the bacterial SynCom used in this work has proved to be a useful tool to promote *M. sativa* growth and nodulation in estuarine soils with moderate to high levels of metal contamination under different abiotic stress conditions (salinity, drought and high temperature). SynCom induction of plant antioxidant enzymatic activities seems to be an important mechanism of plant protection against abiotic stress. The low metal translocation to plant shoots suggests that inoculation with the SynCom could be a safe method to promote legume adaptation in metal contaminated soils.

## Figures and Tables

**Figure 1 plants-12-02083-f001:**
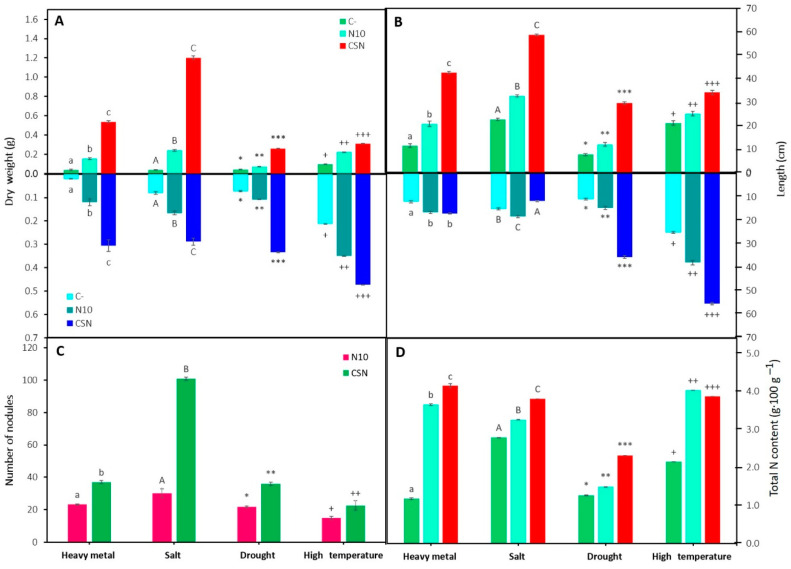
Growth parameters and nodulation. (**A**) Dry weight of shoot (**top** graphs) and roots (**bottom** graphs), (**B**) length of shoot and roots, (**C**) number of nodules, and (**D**) nitrogen content in non-inoculated and inoculated *M. sativa* plants after 60 days in pots under greenhouse conditions using different substrates. Values are means ± S.D. (*n* = 16). Different letters and symbols indicate means that are significantly different from each other (One-way ANOVA, LSD test, *p* < 0.0001). C-: non inoculation; N10: inoculation with *Ensifer* sp. N10; CSN: inoculation with the SynCom formed by *Pseudomonas* sp. N4, *Pseudomonas* sp. N8, *Ensifer* sp. N10, and *Ensifer* sp. N12; heavy metal treatment: soil from the middle marshes of the Odiel river as substrate; salt treatment: sterilized soil from the upper marshes of the Odiel river supplemented with 60 mM of NaCl as substrate; drought treatment: sterilized soil from the upper marshes of the Odiel river as substrate and plants exposed to drought stress; high temperature treatment: sterilized soil from the upper marshes of the Odiel river as substrate and plants exposed to 40 °C.

**Figure 2 plants-12-02083-f002:**
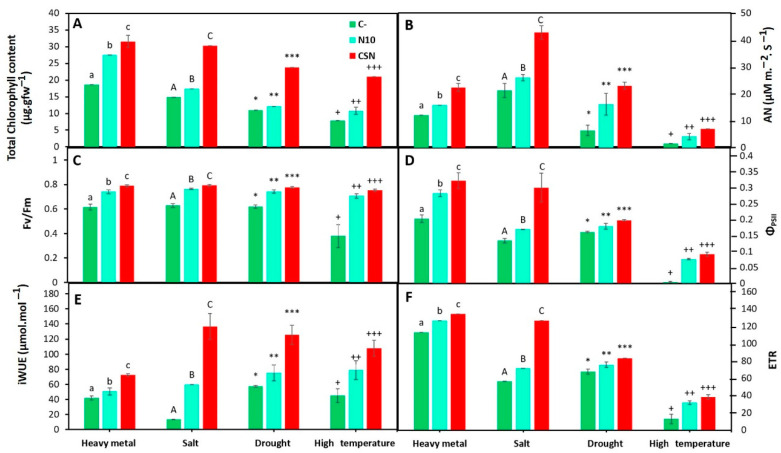
Photosynthetic parameters after the exposition of different stressful conditions. (**A**) Total chlorophyll content, (**B**) net photosynthetic rate (A_N_), (**C**) maximum quantum efficiency of PSII photochemistry (Fv/Fm), (**D**) quantum efficiency of PSII (Φ_PSII_), (**E**) intrinsic water use efficiency (iWUE), and (**F**) electron transport efficiency (ETR) in non-inoculated and inoculated *M. sativa* plants after 60 days in pots under greenhouse conditions in the case of heavy metal and salt treatments, and during the stress in the case of drought and high temperature treatments. Values are means ± S.D. (*n* = 16). Different letters and symbols indicate means that are significantly different from each other. (One-way ANOVA, LSD test, *p* < 0.0001). C-: non inoculation; N10: inoculation with *Ensifer* sp. N10; CSN: inoculation with the SynCom formed by *Pseudomonas* sp. N4, *Pseudomonas* sp. N8, *Ensifer* sp. N10, and *Ensifer* sp. N12; heavy metal treatment: soil from the middle marshes of the Odiel river as substrate; salt treatment: sterilized soil from the upper marshes of the Odiel river supplemented with 60 mM of NaCl as substrate; drought treatment: sterilized soil from the upper marshes of the Odiel river as substrate and plants exposed to drought stress; high temperature substrates: sterilized soil from the upper marshes of the Odiel river as substrate and plants exposed to 40 °C.

**Figure 3 plants-12-02083-f003:**
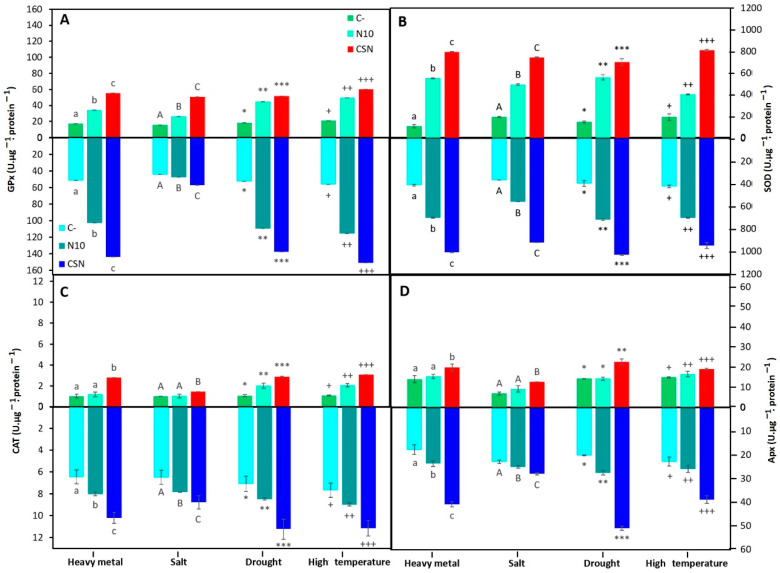
Antioxidant enzymes activities after the exposition of different stressful conditions. (**A**) Guaiacol peroxidase (GPx), (**B**) superoxide dismutase (SOD), (**C**) catalase (CAT), and (**D**) ascorbate peroxidase (GPx) in shoots (top graphs) and roots (bottom graphs) of non-inoculated and inoculated *M. sativa* plants after 60 days in pots under greenhouse conditions in the case of heavy metal and salt treatments, and during the stress in the case of drought and high temperature treatments. Values are means ± S.D. (*n* = 16). Different letters and symbols indicate means that are significantly different from each other. (One-way ANOVA, LSD test, *p* < 0.0001). C-: non inoculation; N10: inoculation with *Ensifer* sp. N10; CSN: inoculation with the SynCom formed by *Pseudomonas* sp. N4, *Pseudomonas* sp. N8, *Ensifer* sp. N10, and *Ensifer* sp. N12; heavy metal treatment: soil from the middle marshes of the Odiel river as substrate; salt treatment: sterilized soil from the upper marshes of the Odiel river supplemented with 60 mM of NaCl as substrate; drought treatment: sterilized soil from the upper marshes of the Odiel river as substrate and plants exposed to drought stress; high temperature substrate: sterilized soil from the upper marshes of the Odiel river as substrate and plants exposed to 40 °C.

**Figure 4 plants-12-02083-f004:**
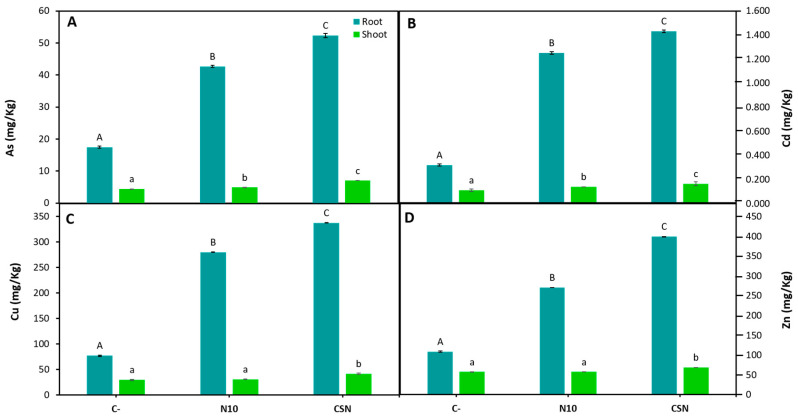
Metal/loids concentration after the exposition of heavy metal stress. (**A**) arsenic (As), (**B**) cadmium (Cd), (**C**) copper (Cu), and (**D**) zinc (Zn) concentrations in shoots and roots of *M. sativa* plants inoculated with different inoculums after 60 days in pots under greenhouse conditions using soil from the middle marshes of the Odiel river as substrate. Values are means ± S.D. (*n* = 16). Different letters indicate means that are significantly different from each other (One-way ANOVA, LSD test, *p* < 0.0001). C-: non inoculation; N10: inoculation with *Ensifer* sp. N10; CSN: inoculation with the SynCom formed by *Pseudomonas* sp. N4, *Pseudomonas* sp. N8, *Ensifer* sp. N10, and *Ensifer* sp. N12.

**Table 1 plants-12-02083-t001:** PGP properties and enzymatic activities in presence of a mix of heavy metals, the maximum tolerable concentration of NaCl and PEG, and the maximum tolerable temperature of the bacterial strains.

PGP Properties	N4	N8	N10	N12
Phosphate solubilization	12	15	12	13
IAA production	3.884	5.729	3.051	4.115
Biofilm formation	0.132	0.104	0.102	0.149
ACC deaminase activity	29.634	19.117	15.388	16.571
N fixation	+	+	–	–
**Enzymatic activities**				
DNAse	–	–	+	+
Amylase	–	–	–	–
Cellulase	–	–	–	–
Lipase	–	–	–	–
Pectinase	+	+	+	+
Protease	–	–	–	–
Chitinase	–	–	–	–
**Maximum Tolerable** **Concentration**				
NaCl	1.5	1.5	2	2
PEG	30	25	30	30
**Maximum temperature**	40 °C	40 °C	45 °C	45 °C

“+”: presence of the activity; “–”: absence of the activity. Values of phosphate solubilization and siderophores production express the diameter of the halo in mm. Values of IAA production are expressed in mg·L^−1^. Values of ACC deaminase activity are expressed in µmoles α-ketobutyrate·mg protein^−1^·h ^−1^). Values of maximum tolerable concentration of NaCl and PEG are expressed in M and percentage, respectively. N4: *Pseudomonas* sp. N4, N8: *Pseudomonas* sp. N8, N10: *Ensifer* sp. N10, and N12: *Ensifer* sp. N12.

**Table 2 plants-12-02083-t002:** Physicochemical properties and micronutrients concentrations of soil from Rio Odiel estuary.

**Physicochemical properties**				
Location	Texture (%) *	Organic material (%)	Conductivity (μS·cm^−1^)	pH
High marsh	72/13/15	0.90 ± 0.01	13.1 ± 0.3	6.8 ± 0.2
Middle marsh	61/21/18	1.59 ± 0.05	13.7 ± 0.2	6.8 ± 0.1
**Metal/loids concentration (mg·kg^−1^)**				
**Location**	**As**	**Cd**	**Cu**	**Zn**
High marsh	27.9 ± 2.2	0.38 ± 0.01	316.7 ± 3.2	345.0 ± 7.1
Middle marsh	162.4 ± 5.6	0.86 ± 0.03	537.5 ± 6.2	865.2 ± 5.6

Values are mean ±S.D. (n = 3). * Texture (sand/slit/clay percentage).

## Data Availability

Not applicable.

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
