# Peer review of "Nodule Synthetic Bacterial Community as Legume Biofertilizer under Abiotic Stress in Estuarine Soils"

_plants, 2023, doi:10.3390/plants12112083_

Round 1

Reviewer 1 Report

The work of the authors seems to be good. It is an opinion and lacks context. More information is needed to provide a meaningful results.

Overall, the abstract presents a study on the potential of a nodule synthetic bacterial community (SynCom) to promote the growth and nodulation of Medicago sativa in degraded estuarine soils under different abiotic stresses. The study found that the plant growth promoting (PGP) endophytes in the SynCom were able to maintain and even increase their PGP properties in the presence of metals, and inoculation with the SynCom enhanced plant growth parameters, nodulation, photosynthesis, and nitrogen content under all conditions tested.

The increase in plant antioxidant enzymatic activities induced by the SynCom appears to be a common and important mechanism of plant protection under abiotic stress conditions. The SynCom was also found to increase M. sativa metal accumulation in roots, with low levels of metals translocation to shoots.

The study suggests that the SynCom used in this work is a safe and effective tool to improve Medicago growth and adaptation to degraded estuarine soils under climate change conditions. However, it would be helpful to include more specific information on the experimental design and results, such as the levels of abiotic stresses tested and the magnitude of the observed effects.

  • "Is that your publication https://www.frontiersin.org/articles/10.3389/fmicb.2022.1005458/full?"

  • "Have the authors matched some results of the published data, or what is the difference between your MS and the previously published MS?"

  • "Why did the authors water the plants with sterile water when, in reality, the soil they obtained did not receive that hygienic treatment?"

  • "Why did the authors focus only on Medicago plants and not try another sustainable plant like mahogany or other plants used to be planted in that contaminated area? Medicago plants could be eaten by animals and could affect their health if they consume plants that have absorbed heavy metals."

  • "Why did the authors mix the obtained soil with perlite?"

  • "The consortium of bacteria used could be difficult to work with together, so how do you convert that quorum into a commercial product to sell?"

  • "The materials and methods section seems to need more information."

Author Response

Thanks a lot for your comments. We have tried to answer all the questions.

  • "Is that your publication https://www.frontiersin.org/articles/10.3389/fmicb.2022.1005458/full?"

            Yes, that is the work describing the isolation and characterization of the SynCom used in the submitted MS. We demonstrated in this work the convenience and advantages of using the four strains for inoculation.

  • "Have the authors matched some results of the published data, or what is the difference between your MS and the previously published MS?"

            In the previous work we tested the effect of plant inoculation with the SynCom, individual strains or couples of strains in the same soil used for bacteria isolation. In the submitted work we have demonstrated the utility of plant inoculation with the SynCom in four conditions, different for the conditions previously assayed, those are, a soil with higher metal contamination, the soil supplemented with NaCl, drought and high temperature. There are coincidences in the results since the SynCom ameliorated plant growth and nodulation under the different conditions and we recorded always the same parameters. We have demonstrated in this MS that the designed SynCom is useful to revegetate areas of the estuary that actually do not have legumes probably as a consequence of salt or heavy metal contamination, and also will be useful to inoculate legumes with our SynCom in a climate change scenario, with longer periods of drought and high temperatures.

  • "Why did the authors water the plants with sterile water when, in reality, the soil they obtained did not receive that hygienic treatment?"

            Indeed, the soils were also sterilized. We wanted to show that the observed effect is only due to inoculation our SynCom or the single inoculation with N10 strain. The water was then sterile too. This fact is reflected in Materials section.

  • "Why did the authors focus only on Medicago plants and not try another sustainable plant like mahogany or other plants used to be planted in that contaminated area? Medicago plants could be eaten by animals and could affect their health if they consume plants that have absorbed heavy metals."

            In this work we focused in Medicago sativa as a model of the Medicago genus and a model legume. Our interest was focused in legumes due to the advantages of these plants as cover crops and their ability to increase soil fertility through nitrogen fixation. In previous works we used a variety of native halophytes and their associated bacteria with phytostabilization potential to revegetate and recover these estuarine soils, see for example the literature cited in the MS, references 11 to 15. We think that legumes could be a good complement to these halophytes in our purpose of soil regeneration. A brief justification on the use of M. sativa has been included in the discussion section.

            Concerning metals accumulation, we and other groups have reported in several works (including results of this MS) that legumes are phytostabilizers of metals, that keep them on the roots, with low metal translocation to shoots. Amounts of metals found on shoots are under the levels allowed for animal or human consumption so they are not a problem for the trophic chain. In the MS we discuss this point. For us, from 2008 in our first paper on effect of As in nodulation, this has been a main concern, to show that legumes retain metals in roots. 

  • "Why did the authors mix the obtained soil with perlite?"

            We add some perlite in order to get some soil aeration and/or humidity retention. This could be useful particularly in drought experiments or under 40ºC, but we use perlite in all our experiments to keep the same conditions in all the situations.

  • "The consortium of bacteria used could be difficult to work with together, so how do you convert that quorum into a commercial product to sell?"

            It looks that the scientific community has now clear the advantages of using a consortium or SynCom with more than one bacteria in this kind of experiments. Concerning commercialization, companies begin to see also the advantages and are preparing products based on two to four strains. We are actually collaborating with several enterprises, as important in Spain as Fertiberia. We have consortiums with three-four bacteria almost ready for commercialization, with these enterprises working in the final formulation of the product. We are not expertise in this final step of the process, but it is possible and they will be frequent quite soon.

  • "The materials and methods section seems to need more information."

We have tried to improve the Materials section, but without knowing exactly which part requires more information is not easy to satisfy the reviewer.

Reviewer 2 Report

The present manuscript “Nodule synthetic bacterial community as legume biofertilizer under abiotic stress in estuarine soils” deals with a highly relevant and innovative topic, which is of high interest for a broad readership.

The Abstract is informative, but could be improved by adding information on the dimension of the effects (% change to the control). Furthermore, it should be stated already in the Abstract that these are results of pot experiments under controlled environmental conditions. The conclusion should be revised to prevent over-interpreting of pot experiments. The conclusion should be reasonably careful. It cannot be stated that the use of this treatment is “a safe tool”, since no eco-toxicological results were presented. It could e.g. be concluded to be a “promising” tool rather.

Conclusion:

Lines 503 – 505: The following sentence has to be deleted: “This SynCom is also an appropriate tool to fight against climate change in these ecologically important ecosystems, protecting legumes from drought and high temperatures and promoting their adaptation.”, since this was not tested in the present study and cannot be stated from pot experiments only!

Author Response

Thanks a lot for the suggestions. We have been more cautious with the conclusions and modified the abstract.

The Abstract is informative, but could be improved by adding information on the dimension of the effects (% change to the control). Furthermore, it should be stated already in the Abstract that these are results of pot experiments under controlled environmental conditions.

            We have modified the abstract to reflect both that are pot experiments and the dimension of effects, unless in a general way, since there are 4 stress conditions and a lot of parameters recorded.

 The conclusion should be revised to prevent over-interpreting of pot experiments. The conclusion should be reasonably careful. It cannot be stated that the use of this treatment is “a safe tool”, since no eco-toxicological results were presented. It could e.g. be concluded to be a “promising” tool rather.

            This suggestion has been followed, see conclusions section.

Reviewer 3 Report

See attached .pdf.

In general, the English is pretty good. It is best in the introduction, discussion, and methods. The results section needs more work.

Author Response

Please, see attached document

Round 2

Reviewer 1 Report

I think the authors review their MS well